# Controlled Drug Delivery Systems for Oral Cancer Treatment—Current Status and Future Perspectives

**DOI:** 10.3390/pharmaceutics11070302

**Published:** 2019-06-30

**Authors:** Farinaz Ketabat, Meenakshi Pundir, Fatemeh Mohabatpour, Liubov Lobanova, Sotirios Koutsopoulos, Lubomir Hadjiiski, Xiongbiao Chen, Petros Papagerakis, Silvana Papagerakis

**Affiliations:** 1Laboratory of Oral, Head and Neck Cancer—Personalized Diagnostics and Therapeutics, Department of Surgery—Division of Head and Neck Surgery, College of Medicine, University of Saskatchewan, Saskatoon, SK S7N 5E5, Canada; 2Laboratory of Precision Oral Health and Chronobiology, College of Dentistry, University of Saskatchewan, Saskatoon, SK S7N 5E4, Canada; 3Division of Biomedical Engineering, University of Saskatchewan, Saskatoon, SK S7K 5A9, Canada; 4Center for Biomedical Engineering, Massachusetts Institute of Technology, Cambridge, MA 02139, USA; 5Departmnet of Radiology, School of Medicine, University of Michigan, Ann Arbor, MI 48109, USA; 6Department of Mechanical Engineering, University of Saskatchewan, Saskatoon, SK S7K 5A9, Canada; 7Department of Otolaryngology-Head and Neck Surgery, School of Medicine, University of Michigan, Ann Arbor, MI 48109, USA

**Keywords:** oral, head and neck squamous cell carcinoma, targeted therapies, drug delivery systems, nanoparticles, controlled drug delivery, circadian clock, chronotherapy, precision medicine

## Abstract

Oral squamous cell carcinoma (OSCC), which encompasses the oral cavity-derived malignancies, is a devastating disease causing substantial morbidity and mortality in both men and women. It is the most common subtype of the head and neck squamous cell carcinoma (HNSCC), which is ranked the sixth most common malignancy worldwide. Despite promising advancements in the conventional therapeutic approaches currently available for patients with oral cancer, many drawbacks are still to be addressed; surgical resection leads to permanent disfigurement, altered sense of self and debilitating physiological consequences, while chemo- and radio-therapies result in significant toxicities, all affecting patient wellbeing and quality of life. Thus, the development of novel therapeutic approaches or modifications of current strategies is paramount to improve individual health outcomes and survival, while early tumour detection remains a priority and significant challenge. In recent years, drug delivery systems and chronotherapy have been developed as alternative methods aiming to enhance the benefits of the current anticancer therapies, while minimizing their undesirable toxic effects on the healthy non-cancerous cells. Targeted drug delivery systems have the potential to increase drug bioavailability and bio-distribution at the site of the primary tumour. This review confers current knowledge on the diverse drug delivery methods, potential carriers (e.g., polymeric, inorganic, and combinational nanoparticles; nanolipids; hydrogels; exosomes) and anticancer targeted approaches for oral squamous cell carcinoma treatment, with an emphasis on their clinical relevance in the era of precision medicine, circadian chronobiology and patient-centred health care.

## 1. Introduction

Oral cancer refers to tumors developed in the lips, hard palate, upper and lower alveolar ridges, anterior two-thirds of the tongue, sublingual area, buccal mucosa, retromolar trigons, and floor of the mouth [1]. The majority (>90%) of oral cancer are carcinomas with squamous differentiation arising from the mucosal epithelium, thus called oral squamous cell carcinomas (OSCCs) [2,3]. In 2018, 354,864 new cases of lip and oral cavity cancer were identified, and 177,384 people died from these types of cancer worldwide [4]. According to the Canadian Cancer Society and the Canadian Dental Association, the incidence of OSCC has increased in Canada in both males and females since mid-1990s; 4700 new cases of oral cancer and 1250 oral cancer-related deaths were reported in Canada in 2017 alone [5,6]. Most often diagnosed at late stages (approximately 60% of patients present with advanced stage disease at the initial diagnosis, OSCC remains one of the most difficult challenges in head and neck oncology, and continues to be a disfiguring and deadly disease with dismal 50% to 60% five-year disease specific survival rate [7,8]. Due to its anatomic location, OSCC progression and treatment significantly impact patient quality of life, involving impairment of most vital functions (e.g., speech, swallowing, taste.), appearance and sense of self; they are associated with profound functional morbidity even when the cancer is cured [3,9]. 

New trends have recently emerged in the OSCC patient profile including younger patients (younger than 50 years), particularly those with human papillomavirus (HPV)-positive tumors [10,11]; a steady change in the OSCC sex ratio with a worrisome increase in OSCC incidence and mortality in females [12]; and the implications of novel, previously unrecognized factors, such as the circadian clock disruption in the initiation and progression of the OSCC [13,14,15,16].

OSCC has traditionally been associated with risk factors such as tobacco and alcohol consumption; however, HPV, a well-known cause of cervical cancer, has emerged in recent years as an etiological cause for a subset of head and neck squamous cell carcinoma (HNSCC), particularly in patients who lack the traditional risk factors [17,18]. The majority (60–80%) of HPV-driven cancers of the head and neck are oropharyngeal squamous cell carcinomas (comprising the tonsils and the base of the tongue). Recent studies have identified various types of HPV associated with both benign and malignant lesions in the oral cavity [19,20,21,22].

The HPV diagnosis is critical in planning treatment for oropharyngeal cancer (OPC) patients [23,24,25]. Within OPC, there is a marked difference between clinical behaviors and outcomes for patients who test positive versus negative for HPV infection. For high/late-stage patients, HPV positivity has become a significant prognostic factor that is critical for guiding the choice of treatment, with an HPV positive diagnosis resulting in lower toxicities and improved outcomes [26]. In contrast, a significant subset of early-stage OPC patients are HPV negative, their cancer rapidly progresses into advanced metastatic tumors and fails to respond to the standard of care with poor outcomes and survival. Patients with chronic exposure of the entire mucosa of the upper digestive tract (cancerization field) to carcinogenic factors (e.g., from tobacco, alcohol, and betel quid chewing) are at a higher risk for multiple primary tumors.

Oral squamous cell carcinoma is the second most common cancer in transplant patients (e.g., treated for leukemia, lymphoma, multiple myeloma, etc) [27]. The conventional approaches for oral cancer treatment involve surgery, which is the treatment of choice, ionizing radiation which is the prevalent non-surgical therapeutic approach, or a combination of radio-, chemotherapy, and surgery [28]; surgical resection leads to permanent disfigurement, altered sense of self and debilitating physiological consequences, substantial functional impairment, and morbidity, while chemo- and radio-therapies result in significant toxicities, all affecting patient wellbeing and quality of life. These treatments are efficient for the treatment of the primary tumor but are used with palliative intent in advanced cases with metastatic disease, with significant side and adverse effects [29]. Despite the advances in surgery, chemotherapy, and radiotherapy for HNSCC treatment, the prognosis for this disease has not been significantly improved over the last 50 years [8]. Thus, the development of novel therapeutic approaches or modifications of current strategies is paramount to improve individual health outcomes and survival, while early tumor detection remains a priority and significant challenge.

The oral, head, and neck cancer is an immunosuppressive disease (characterized by a lower absolute lymphocyte count and poor antigen-presenting function) that interferes with the patient’s natural immune response, preventing tumor cell recognition and immune-mediated clearance [30]. Immunotherapy, a recently developed cancer treatment modality, has shown promise as an additional therapeutic option in patients having failed multiple prior therapeutic modalities, due to the success of immune-modulating agents in patients with refractory solid tumors [31,32]. The goal of immunotherapy as an anticancer approach is to either block the pathways cancer cells use to escape the immune system or to enhance the patient’s immune reactions directed against tumor cells [33]. Anti-cancer immunotherapy includes: (1) systemic therapy, which is a systemic immune activation including administration of systemic cytokine, cancer vaccines, or adoptive cell transfer; (2) local-based therapy, which is based on changes in local immune status including modulation of the immunosuppressive tumor microenvironment, with immune checkpoint or small molecular inhibitors [34]. Immune-modulating approaches available for the treatment of head and neck cancer target a variety of immune processes and critical checkpoints, including cytotoxic T-lymphocyte associated antigen-4 (CTLA4), and program death (PD-1) and its ligand (PD-L1); other methods using immune modulating molecules as well as combinatorial trials evaluating these agents in the first-line setting and early-stage disease are under development [35,36]. 

Because HNSCC tumors have been shown to poorly present tumor antigen (TA) on the cell surface, monoclonal antibodies facilitating better TA presentation are one avenue for targeted therapeutics [37]. Nivolumab and pembrolizumab, two anti-PD-1 agents, recently approved for use as monotherapy in the second-line setting for patients with platinum-refractory recurrent/metastatic HNSCC, have shown efficacy in clinical trials [30,38]. Other targeted therapies using epidermal growth factor receptors (EGFR, highly overexpressed in 80–90% of HNSCC) inhibitors, such as cetuximab, bevacizumab, and erlotinib, have shown improvement of OSCC patient survival [39]. Despite the promise of immunotherapies, new therapeutic approaches or improvements to clinical trials design that are tailored on the tumor/patient profile are much needed in order to overcome the innate and acquired tumor resistance, as well as to address/prevent their side and adverse effects [40]. Developing novel immunotherapeutic approaches can be promising in providing long-term control of the disease in the response population, although the low efficacy and high toxicity in some patients can be a severe issue [33,34,38]. Generally, using immunotherapy can be challenging due to auto-immune side effects, variability in tumor responses rate, and financial cost [36]. A solution to enhance the efficacy of immune agents is using nano-based drug delivery systems (DDSs) through direct targeting of the cancer cells, facilitating intracellular penetration, and boosting the immunogenicity of antigens [41]. To date, there are limited studies on the utilization of DDS combined with immunotherapy for the treatment of HNSCC or OSCC. Hirabayashi et al. and Maeda et al. developed anti-EGFR antibody-conjugated microbubbles for the treatment of HNSCC and OSCC, respectively [42,43]. These studies showed promising results for future applications of combined immunotherapy with DDSs.

DDS have been developed as an alternative method aiming to enhance the benefits of the current anticancer therapies, while minimizing their undesirable toxic effects on the healthy cells. For instance, the chemotherapeutic agents have several limitations in terms of oral bioavailability, stability in natural conditions, and non-specific bio-distribution, that decrease their therapeutic efficiency [44,45]; their side effects can be severe particularly in older patients with debilitating comorbidities. For instance, the parenteral administration of chemotherapeutic drugs allows for the drug control via the bloodstream, thus affecting other non-cancerous organs/tissues in the body, besides the tumor itself; the extent and clinical consequences of these non-specific effects are hard to predict. Adverse effects such as nausea, vomiting, hair loss, infections, and diarrhea are common in patients receiving chemotherapy. Radiotherapy can be used alone or in combination with the chemotherapy to treat the primary tumor; shrink the tumor prior to surgery (neoadjuvant therapy; note: chemotherapy also can be administered in the neoadjuvant setting); as adjuvant therapy to maximize the effectiveness of the primary treatment in hopes of extending survival and reducing the risk for recurrence; or to relieve pain or control symptoms of advanced oral cancer (palliative therapy). A patient’s response to neoadjuvant therapy can determine which adjuvant therapy is selected. Side effects of radiation therapy due to transient or permanent damage to healthy tissues are fatigue, sore or dry mouth and difficulty swallowing, dental problems (tooth decay), taste change, loss of appetite, nausea and vomiting, nerve damage, pain, infection, osteoradionecrosis, trismus, lymphedema, and hair loss [29]. These can affect the ability to eat and speak and can lead to other complications such as dehydration and malnutrition, social withdrawal, anxiety and depression, impacting the patient’s quality of life.

Conventional therapeutic approaches need improvement in bioavailability and targeted delivery to the tumor site (for a pre-determined period) to overcome and prevent the adverse side effects of the drugs [46]. Our group has investigated the potential anticancer benefits of antacid medications, such as proton pump inhibitors and histamine 2 blockers that are commonly used in HNSCC patients to manage acid reflux, a condition that contributes to complications after surgery or during radiotherapy. Our findings in a large cohort study indicated that routine clinical usage of these two classes of antacids in HNSCC patients was correlated with enhanced survival; remarkably our analysis identified histamine 2 receptor antagonist class usage as a significant prognostic factor for recurrence-free survival in patients with oropharyngeal tumors HPV-positive [47]. Ongoing studies in our laboratory are investigating the abilities of these medications to improve the efficacy of conventional therapies, particularly in advanced HNSCC [48,49]. 

An innovative approach to improve the efficacy of chemotherapeutic agents is the administration of drugs in a time-specific manner (chrono-chemotherapy). It is becoming evident that administration timing is as vital as the dosing amount of chemotherapy [50]. The time of administration (morning vs. evening) influences drug toxicity and therapeutic efficacy because human body physiology is affected by the circadian clock rhythms [51]. Anticancer chemotherapeutic agents docetaxel, doxorubicin, fluorouracil, and paclitaxel have been recently recognized by the World Health Organization as drugs which target circadian clock genes (Bcl2, Top2a, Tyms, and Bcl2 respectively). Hence, they can be employed in chrono-chemotherapy for oral cancer treatment [52]. A recent study showed that chrono-chemotherapy of a combination of Docetaxel, Cisplatin, and Fluorouracil (DCF) helped to decrease the severity of the side effects of each of these drugs [53]; patients with OSCC had less vomiting, nausea, and neutropenia when treated with evening DCF dosing rather than with morning administration [53]. Thus, it seems promising that chrono-chemotherapy has the ability to reduce the severity/extent of the side effects of some chemotherapeutic drugs, which can be exploited as a novel therapeutic strategy in oral, head and neck cancer patients and beyond.

Another approach that showed promise in overcoming the complications of conventional anticancer agents while enhancing their therapeutic efficacy is the targeted drug delivery system consisting of natural and/or synthetic polymers for delivery of chemotherapeutic agents to the tumor site. Targeted drug delivery systems have the potential to increase drug bioavailability and bio-distribution at the site of the primary tumor. DDS is capable of releasing a bioactive molecule at a specific site with a specific delivery rate. Targeted DDS for oral cancer could thus improve patient compliance, enhance drug efficiency while reducing treatment duration, and consequently decrease healthcare expenses. In vivo studies have shown that targeted DDS can also improve the half-time of otherwise rapidly degradable drugs such as peptides and proteins, thus prolonging their local effects [54].

Our review of the most promising anticancer drug delivery approaches is structured in three sections as follows: first, the conventional anticancer drugs are reviewed in regard to their oral administration and potential for DDS formulation; second, a brief background of commonly used carriers in DDS for oral cancer treatment is provided; and third, the potential of different drug delivery methods for OSCC is discussed.

## 2. Anticancer Agents for Oral Cancer Treatment Formulated in Drug Delivery Systems

While most of the oncological treatments are traditionally administered intravenously, several anticancer drugs have recently been developed and approved by the USA Federal Drug Administration (FDA) for oral administration [55]. 

Administration of chemotherapeutic drugs in the form of pill or gel is an attractive approach to enhance patient compliance. This method of delivery is also desirable when the treatment requires drug exposure for prolonged periods [46]. Unfortunately, oral administration of most anticancer drugs is hindered due to the drug’s physicochemical characteristics, particularly poor aqueous solubility [56,57]. However, most of the chemotherapeutic agents delivered intravenously can also be administered via other routes of delivery when incorporated in suitable carrier (bio)materials [58]. Carefully designed DDS can be used to formulate chemotherapeutic agents for local (e.g., applied to the tumor site) or intravenous delivery with higher efficacy than the standard intravenous administration. Following is an overview of the most common anticancer drugs used for the treatment of oral cavity and oropharynx cancer patients [59], which have already been investigated for their administration using controlled and/or targeted DDS with promising results.

### 2.1. Paclitaxel (PTX)

Paclitaxel (Taxol) is an antineoplastic agent which functions by cellular growth inhibition. Oral administration of PTX is challenging because of its low solubility and reduced permeability across the intestinal epithelium/mucosa that limit its absorption. When PTX is administered intravenously, which is the most common delivery method in the clinic, its distribution throughout the body is very extensive, causing severe side effects such as liver dysfunction [60]. To increase its absorption, Lee et al. designed a platform based on the chemical conjugation of PTX to the low molecular weight chitosan, which increased PTX’s water solubility due to the presence of chitosan and its increased retention time in the gastrointestinal (GI) tract [61]. Tiwari and Amiji reported nano-emulsion formulations of PTX to improve its oral bioavailability; the nano-emulsion delivery of PTX resulted in a significant increase of the PTX concentration in systemic circulation versus control (aqueous solution of PTX), suggesting that this formulation can enhance the oral bioavailability of hydrophobic drugs such as PTX [62]. In another study, Dong and Feng added montmorillonite to poly(lactic-*co*-glycolic acid) (PLGA) in order to synthesize nanoparticles for PTX delivery; the montmorillonite-PLGA nanoparticles allowed for an enhanced cellular uptake and efficiency of PTX as compared to the PLGA nanoparticles alone, suggesting that the montmorillonite-PLGA nanoparticle formulation can extend the residence time of PTX in the GI tract [63].

### 2.2. Cisplatin (DDP)

Cisplatin is a chemotherapeutic agent with a recognized benefit in the treatment of various human cancers, including oral, head and neck squamous cell carcinoma, bladder, lung, ovarian, breast, and testicular cancers. Cisplatin causes apoptosis (cell death) of cancer cells due to its ability to crosslink with purine bases on DNA, interfering with DNA repair mechanism, and causing DNA damage [64,65]. Because its administration has been associated with severe side effects such as renal failure, there have been several attempts to formulate this drug in an oral sustained release system [64]. Cheng et al. exploited the ability of the low pH-responsive porous hollow nanoparticles of Fe_3_O_4_ to be used as a vehicle for site-specific cisplatin delivery; their system, based on the encapsulated cisplatin into porous hollow nanoparticles of Fe_3_O_4_, not only protected cisplatin from deactivation by plasma proteins and other biomolecules before reaching the target site, but also provided control of the release rate of cisplatin by varying the nanoparticle’s pore size and pH [66]. Yan and Gemeinhart generated encapsulated cisplatin poly(acrylic acid-*co*-methyl methacrylate) micro-particles for controlled release of cisplatin, and their system enabled cisplatin to maintain its activity for prolonged periods [67]. A cisplatin analog with similar chemotherapeutic profile, Carboplatin, has also been investigated alone or as part of nanoparticle formulations in order to minimize its undesired side effects [68].

### 2.3. Doxorubicin

Doxorubicin (DOX) is one of the most potent anticancer agents used for the treatment of numerous cancer types, because of its ability to target rapidly dividing cells, both cancerous and non-cancerous. Its toxicity on non-cancerous cells limits its application because it can result in cell death in major organs such as heart, brain, liver, and kidney [69,70,71]. Drug delivery strategies sought to minimize DOX side effects while exploiting its anticancer properties with higher therapeutic efficiency. For instance, Li et al. encapsulated DOX in dextran nanoparticles to specifically target tumor cells with the expectation that these smart nanoparticles would increase drug loading efficiency and release the drug at a particular site directly into the cancer cell’s nucleus [72]. She et al. used dendronized heparin nanoparticles conjugated to DOX as a pH-responsive drug delivery vehicle for cancer treatment. These nanoparticles showed significant anti-tumor activity on a 4T1 breast tumor model without toxicity to healthy organs [73]. Collectively, this evidence showed that incorporating DOX into nanoparticles held promise for reducing toxicity on healthy cells while increasing its antitumor activity.

### 2.4. Docetaxel

Docetaxel (DTX), an effective anticancer drug, is most commonly administered intravenously in cancer patients because of its highly hydrophobic property, but it has low oral bioavailability due to the P-glycoprotein (P-gp)-mediated efflux and first passes effect. To address these drawbacks, Sohail et al. synthesized a chitosan scaffold in which folic acid and thiol groups were grafted to chitosan to target cancer cells and improve permeation through the gastrointestinal tract [74]. They also synthesized silver nanoclusters in situ, which allowed for the generation of core-shell nano-capsules with the hydrophobic DTX as the core and the silver nanocluster embedded chitosan as the shell; this strategy resulted in a DTX carrier system suitable for the oral delivery of DTX to cancerous tissues [30].

### 2.5. Methotrexate

Methotrexate (MTX), an antimetabolite agent used in anticancer chemotherapy, is a folate antagonist which inhibits the synthesis of purines and pyrimidines, thereby causing inhibition of the malignant cells’ proliferation. MTX is used for the treatment of a variety of cancers, including oral, head, and neck cancer, acute lymphocytic leukemia, non-Hodgkin’s lymphoma, choriocarcinoma, osteosarcoma, and breast cancer [75,76]. When administered orally, MTX systemic bioavailability is approximately 35%, which is significantly lower than when administered parenterally [77]. Oral administration of MTX is associated with significant side effects (diarrhea, ulcerative stomatitis, hemorrhagic enteritis, gastrointestinal perforation) due to inhibition of cellular proliferation. Kumar and Rao formulated MTX in proteinoid microspheres to enhance its bioavailability and targetability, with the expectation that these microspheres could deliver MTX and other pharmaceutical compounds that are prone to degradation, under gastric condition [78]. Paliwal R et al. encapsulated MTX into solid lipid nanoparticles (SLNs) consisting of stearic acid, glycerol monostearate, tristearin, and Compritol 888 ATO; the MTX loaded SLNs significantly improved the bioavailability of MTX by protecting MTX from degradation in the harsh gastric conditions [79].

### 2.6. Fluoropyrimidine 5-Fluorouracil

Fluoropyrimidine 5-fluorouracil (5-FU), another FDA approved anticancer drug, inhibits essential biosynthesis processes or interferes with DNA or RNA, limiting their normal function. This drug has been effective in treating various types of cancer, including oral, head and neck cancer, colorectal, and breast cancer [80]. Li et al. designed a biodegradable controlled release system composed of PLGA nanoparticles, which maintained a prolonged continuous release of 5-FU. Their results showed that these nanoparticles could enhance the oral bioavailability of 5-FU while decreasing its local gastrointestinal side effects [81]. Minhas et al. developed a pH-responsive controlled release system for 5-FU delivery, by preparing a chemically cross-linked polyvinyl alcohol-*co*-poly(methacrylic acid) hydrogel loaded with 5-FU, which enabled the release of 5-FU at pH 7.4, with the potential for being used as an oral drug delivery vehicle for 5-FU in cancer treatment, particularly colorectal cancer [38].

## 3. Carriers for OSCC Drug Delivery Systems

Carrier-based drug delivery systems are used for controlled release of drugs while providing improved selectivity and effectiveness, and reduced side effects compared to the chemotherapeutic agents alone. Different carrier systems based on nanoparticles, nanolipids, and hydrogels are discussed here, each with unique advantages and disadvantages (Figure 1). Additionally, exosomes have been recently introduced as potential carriers of chemotherapeutic agents for oral cancer treatment. The benefits and drawbacks of each carrier system are summarized in Table 1.

### 3.1. Nanoparticles for Drug Delivery

The use of nanotechnology in drug delivery has allowed for selective and safe methodologies for OSCC treatment [87,92]. Nanoparticles provide enhanced bioactivity due to their large surface to volume ratio [84,96]. The most common nanoparticles investigated in oral cancer treatments include gold nanoparticles, liposomes, magnetic nanoparticles, and polymeric micelles [88,97]. These nanoparticles are capable of killing cancer cells by delivering the drugs entrapped or encapsulated in them [92,97]. Utilizing nanoparticles as drug carriers have also resulted in stabilization of chemotherapeutic compounds that can be released in a controlled and sustained manner. This targeted delivery facilitates the prolonged release of a drug at a specific site, thus reducing its systemic toxicity [98].

#### 3.1.1. Polymeric Nanoparticles for Drug Delivery System

For targeted drug delivery with improved biocompatibility and drug controlled release, nanoparticles fabricated from natural and synthetic polymer have received much attention [84]. Polymers consisting of polysaccharides, poly(lactic acid) (PLA), poly(glycolic acid) (PGA) and their copolymers, are biodegradable and thus slowly eliminated from the body after the delivery of cargo [99]. There has been substantial research into intraoral, site-specific chemoprevention using a polymeric drug delivery system. These chemopreventive agents are delivered directly to various affected sites within the oral cavity, thereby preventing the malignant conversion of oral epithelial dysplasia to frank carcinoma. Several techniques are currently employed to synthesize such nanoparticles, including nanoprecipitation, emulsifications, and self-assembly [100]. Selecting a particular method depends on the physicochemical properties of the polymer, drug solubility, and drug release behavior [100].

Endo et al. have used polymeric nanoparticles based on poly(ethylene glycol)-poly(glutamic acid) block copolymer to increase the anti-tumor effects and reduce the toxicity of cisplatin [101], the most commonly used chemotherapeutic drug in OSCC patients [102]. Cisplatin was integrated into polymeric micelles through the polymer-metal complex formation between poly(ethylene glycol)-poly(glutamic acid) block copolymers and CDDP (NC-6004). The mean particle size of polymeric micelles (NC-6004) was 30 nm. Also, static light scattering (SLS) measurement exhibited that there is no dissociation of cisplatin-loaded micelles upon dilution and the critical micelles concentration (CMC) was less than 5 × 10^−7^ [100,103]. 

The treatment of oral cancer cells with cisplatin-loaded nanoparticles (NC-6004) leads to the activation of the caspase-3 and caspase-7 pathways, which induce apoptosis [101]. In vivo results showed that the antitumor activity of NC-6004 against tumor growth in oral carcinoma-bearing mice was 4.4–6.6-fold higher compared to the control group. Additionally, the controlled release of cisplatin from these nanoparticles resulted in decreased nephrotoxicity and neurotoxicity compared with administration of cisplatin in solution [101].

Additional agents (e.g., curcumin) have been investigated for their therapeutic benefit in oral cancer based on their ability to induce apoptosis and inhibit tumor cell proliferation [91,104]. To enhance the clinical benefits of these therapeutic agents by improving their bioavailability and stability, Mazzarino et al. used a nanoprecipitation technique to generate polycaprolactone (PCL) nanoparticles coated with the polysaccharide chitosan for curcumin delivery into the oral cavity [104]. The chitosan coating on the nanoparticles was confirmed by the changes in particles size and zeta potential measurements. With the increase in concentration of chitosan, the hydrodynamic radius of nanoparticles increased for unloaded and curcumin-loaded nanoparticles (104 to 125 nm; polydispersity index (PDI) < 0.2) [101]. Additionaly, chitosan-coated nanoparticles showed increased zeta potential values (positive surface charge) compared to uncoated nanoparticles due to the presence of positively charged amino groups of chitosan molecules on the surface of the particles, thus proving that the nanoparticles were successfully coated [105]. Also, due to a strong interaction between curcumin and PCL, the core of the curcumin-loaded nanoparticles was compacted, which leads to the decrease in their size compared to the unloaded nanoparticles [101,105].

Adsorption of chitosan on PCL formed a muco-adhesive nanoemulsion, which showed an interaction between glycoprotein mucin and PCL nanoparticles. This system was evaluated by surface plasmon resonance. Better muco-adhesive properties lead to an increase in the residence time of the drug. The cytotoxic effect of these nanoparticles was evaluated in an in vitro study using an OSCC-derived cell line, SCC-9 that showed induction of apoptosis in tumoral cells. Furthermore, these polymeric nanoparticles encapsulating curcumin showed improved bioavailability [104] and improved curcumin stability by preventing its degradation in neutral solutions and upon exposure to light [106].

Interestingly, dietary substances containing bioactive compounds may also have some ability to suppress cancer. Studies indicated that ellagic acid (a polyphenolic chemopreventive agent) has anti-cancerous, antioxidant, and antiviral properties. However, its usage is limited due to low oral bioavailability and water solubility [107]. Bio-polymeric nanoparticles may overcome these drawbacks, increasing the drug efficiency by preventing the degradation of unstable chemotherapeutic biomolecules. Arulmozhi et al. developed chitosan nanoparticles encapsulating ellagic acid using the ionotropic gelation technique, which enhanced the anticancer properties of ellagic acid, thus, making this formulation a promising platform for oral cancer treatment [100,108].

#### 3.1.2. Inorganic Nanoparticles for Drug Delivery System

Inorganic nanoparticles have been extensively used in treatments due to their lower toxicity, higher tolerance towards organic solvents, and better bioavailability compared with the free drug [88]. Inorganic nanoparticles based on noble metals (e.g., gold) have been used in diagnostic and imaging processes and received much attention due to their highly controlled optical properties [109,110]. Such nanoparticles are potential photo-thermal agents with high efficacy in therapeutic applications. Sayed et al. prepared anti-epithelial growth factor receptor (EGFR) antibody-conjugated gold (Au) nanoparticles (with an average particle size of 40 nm characterized by transmission electron microscopy (TEM) and incubated them with OSCC cell lines and a control benign epithelial cell line [110]. Continuous wave (CW) argon ion laser was used to produce photothermal destruction. These in vitro results showed that the malignant cells with anti-EGFR/Au conjugates required less energy to produce photothermal destruction due to the targeting of the Au nanoparticles on the surface of EGFR-overexpressing malignant cells but not on benign cells. In clinical applications, near-infrared (NIR) laser light with deep penetration allowed for effective delivery of anti-EGFR/Au conjugates to the cells. Furthermore, the surface plasmon absorption of Au nanoparticles can be finely tuned by modifying the nanoparticles’ size to allow for better absorption of this NIR laser light, thus maximizing their therapeutic benefit [110].

Recently, other therapeutic techniques, including photodynamic therapy (PDT), have been employed to increase the penetration of drugs deeper into tissues, required for the treatment of advanced and recurrent oral cancer [111]. Lucky et al. developed up-conversion nanoparticles (UCN) loaded with PEGylated titanium dioxide (TiO_2_) to increase tissue penetration using NIR; these nanoparticles were used for targeting EGFRs on the surface of cancer cells using anti-EGFR-antibody conjugated with PEGylated TiO_2_-UCNs to inhibit tumor proliferation, invasion, angiogenesis, and metastasis. Anti-EGFR-PEG-TiO_2_-UCNs nanoparticles were characterized by TEM and a well-defined core-shell structure was observed with approximately 50 nm in diameter. Further, the composition of nanoparticles was confirmed by Energy-dispersive X-ray (EDX) spectroscopy showing formation of Na (Sodium), Y (Yttrium), F (Fluorine), Yb (Ytterbium), and Tm (Thulium) from the core nanocrystals and Ti (Titanium), Si (Silicon) and O (Oxygen) from the shell [107]. In vivo studies investigating anti-EGFR-PEG-TiO_2_-UCNs showed no toxic side effects, whereas in vitro studies showed enhanced apoptosis and tumor growth inhibition [111,112].

Drug delivery using nanoparticles allowed for increased concentration of therapeutic agents at the tumor site, which resulted in cancer cell inhibition with reduced toxicity on the surrounding non-cancerous healthy cells. Nevertheless, there are still challenges linked to carriers stability and fate in the human body, and their limited effective delivery remains problematic. To overcome some of these drawbacks, Eguchi et al. prepared innovative magnetic nanoparticles consisting of μ-oxo *N*,*N*′-bis (salicylidene) ethylenediamine iron (Fe(Salen)) for targeted delivery of anticancer agents. Since these particles were difficult to solubilize, they were suspended in water or saline after sonication. Iron–salen particles were characterized using DLS and TEM, showing size ranged 1.2–3 µm for unsonicated particles and 60–800 nm for sonicated particles. The sonication for approximately 6 hours reduced particle size (confirmed by TEM) with smooth edges of the particles as compared to the unsonicated particles. The sonicated Fe(Salen) particles showed zeta potential value of −24.1 mV, thus confirming the stability of the colloidal dispersion [113].

Sato et al. used Fe(Salen) nanoparticles with average size of 200 nm for targeted delivery of anticancer agents. These nanoparticles were sonicated for 30 min and were suspended in normal saline. Alternating magnetic field (AMF) combining chemotherapy and hyperthermia was used to heat Fe(Salen) nanoparticles and resulted in increased induction of cancer cell apoptosis and better carrier stability, as compared to individual chemotherapy or magnetic guided delivery. Fe(Salen) nanoparticles were useful for controlled drug delivery and hyperthermia therapy, with an increase in anti-cancer therapeutic efficacy and reduced toxicity [89].

Other inorganic nanoparticles systems, such as mesoporous silica nanoparticles (MSNP), showed promise for cancer therapy. These nanoparticles’ advantages include high porosity, biocompatibility, and amenability for surface functionalization [114]. The porous nature of MSNPs provides much free space for antitumor drugs to be incorporated. These nanoparticles, combined with polymers, can carry drugs with high efficiency in targeting OSCC cells [114,115], but additional investigations are required for the routine implementation of these systems into clinical practice.

#### 3.1.3. Combinational (Polymeric-Inorganic) Nanoparticles

Combinational drug treatment is recognized for its increased therapeutic benefits. Targeted drug delivery offers improved therapeutic efficacy with reduced toxicity. Quinacrine (QC) is an anticancer agent that is also used as an antimalarial drug; it has shown therapeutic benefits in breast, lung, colon, and renal cell carcinoma. Despite these positive outcomes, QC clinical applications are limited due to its poor bioavailability and various side effects, including skin rash and pigmentation, and immunological complications [116]. Inorganic silver-based nanoparticles (AgNPs) also have potential as anticancer agents due to their ability to induce tumor cell apoptosis. Combinational approaches have been employed to address AgNP’s limitation of toxicity to healthy cells at higher doses, which resulted in the enhanced anticancer activity of AgNPs [100,116]. Satapathy et al. prepared highly stable PLGA based quinacrine (QC)–silver hybrid nanoparticles (QAgNP) using an oil-in-water emulsion solvent evaporation technique. The TEM analysis determined the size and morphology of QAgNP with size ranging 50–100 nm. Average particle size of 382.4 ± 0.11 nm was obtained by DLS with a positive zeta potential of 0.523 ± 0.09 mV [111]. These nanoparticles were allowed to interact with various oral cancer cell lines and OSCC-derived stem cells and evaluated for their antitumor activity. PLGA/quinacrine/silver nanoparticles showed high cytotoxicity against cancer cells with improved ability to destroy specifically the OSCC-derived stem cells. The study also confirmed that PLGA/quinacrine/silver nanoparticles not only inhibited proliferation of OSCC but also reduced neo-angiogenesis, suggesting that this hybrid nanoparticle drug delivery system can be a promising platform for the treatment of OSCC [100,116].

### 3.2. Nanolipids

Polymeric nanoparticles’ cytotoxicity, due to low internalization into the tumor cells, restricts their therapeutic efficiency [85,86]. Solid lipid-based nanoparticles (SLNs) have overcome this problem because they can penetrate cancer cells. Furthermore, their high stability provides controlled drug release, drug protection from chemical degradation, and they can serve as carriers for drugs with low aqueous solubility [92,117]. Therefore, these nanoparticles seem suitable for local delivery of drugs and chemopreventive agents [118,119].

One limitation of nanoparticles prepared from solid lipids is their crystalline structure, which allows for only limited space to accommodate drugs. Nanostructured lipid carriers (NLCs) have been designed and tested in cancer therapy to overcome this limitation. These NLCs consist of both solid and liquid lipids in a core matrix, thereby distorting the crystal structure and providing space for drugs to be encapsulated in amorphous clusters [120,121]. Thus, NLCs addressed the issues of poor solubility, low bioavailability, and instability of anticancer drugs and therapeutic agents [93,121]. A recent study by Fang et al. reported the enhanced bioavailability of curcumin loaded into nanostructured lipid particles, an emerging method for treating OSCC [122]. Other studies reported the fabrication of nanostructured lipids with other therapeutic agents, such as docetaxel and etoposide, which have shown promise in treating oral cancer [123,124,125].

### 3.3. Hydrogel-Based Drug Delivery Systems

Hydrogels are three-dimensional (3D) mesh structures of hydrophilic fibers that contain a large amount of water or biological fluids. Hydrogels resemble the soft body tissues and are capable of encapsulating drugs and biomolecules such as proteins and genetic materials [126]. Depending on the mechanism used for their gelation, there are two types of hydrogels, physical and chemical. Physical gelation is not inherently permanent, but reversible whereas chemical gelation is reversible because it involves chemical bonds, and thus results in permanent or very stable hydrogels [127,128,129].

Hydrogels act as localized, targeted drug delivery systems and offer some advantages when juxtaposed with active and passive targeting by using nanocarriers [130]. For instance, a limitation of nanoparticle-based systems is the swift elimination from blood circulation due to their small size and renal clearance. Also, the tumor microvascular morphology, characterized by increased interstitial fluid pressure, results in low intra-tumoral penetration of the drug-loaded nanocarriers, which in turn results in decreased therapeutic efficiency [130,131,132,133]. In contrast, hydrogels can provide sustained administration of both hydrophilic and hydrophobic drugs, proteins and other biomolecules independently of the microvascular system of the tumor, allowing for high drug loading capacity, as high as the drug’s solubility in water [134,135]. Hydrogels can also control the release of the drug for short or long periods (up to several months) by altering the density of the nanofibers in the hydrogel [136].

Moreover, hydrogels allow for co-administration of multiple drugs with synergistic anti-cancer effects and decreased drug resistance [46,130]. In one study, a thermosensitive physical hydrogel composed of poly(ethylene glycol)-poly(ε-caprolactone)-poly(ethylene glycol) (PEG-PCL-PEG, PECE) showed great potential as an in situ controlled delivery system for suberoylanilide hydroxamic acid (SAHA), a histone deacetylase (HDAC) inhibitor in combination with cisplatin (DDP). When injected intratumorally in a OSCC mouse model, the PECE hydrogel provided sustained release of the loaded SAHA and DDP for more than 14 days, enhanced therapeutic effects, and reduced side effects [137].

### 3.4. Exosomes

Exosomes are membranous vesicles with sizes between 40–120 nm that are secreted by different cells, such as dendritic cells, macrophages, mesenchymal stem cells, endothelial, and epithelial cells, into the extracellular space [138,139,140,141]. Due to their nanosized dimensions and natural formation, exosomes have received much attention and are involved in many biological and pathological processes. Exosomes are secreted when the multivesicular body (MVB) fuses with the plasma membrane. Exosomes can contain many types of biomolecules and play an essential role in inter-cellular communication [142]. Their ability to bind to the cell membrane through adhesion proteins and ligands has made them a sound carrier system for targeted drug delivery applications [139,141]. They have been used as a vehicle for chemotherapeutic agents such as curcumin, DOX, and PTX, helping to reduce their side effects while increasing their therapeutic efficiency [139,143,144]. Tian et al. used targeted exosomes as a targeted delivery system for DOX to treat breast cancer cells; when injected intravenously in mice, these exosomes delivered DOX targeted to tumor tissues, which resulted in inhibition of tumor growth without overt toxicity [144]. Despite their promising preclinical evidence for cancer therapy, several limitations prevent exosomes utilization as an efficient drug delivery system in the clinical practice, mainly due to their limited capacity to deliver high doses of therapeutic agents. Also, the separation of exosomes with high purity is a long and demanding process that usually generate low amounts. Finally, studies showed that exosome administration in patients might lead to adverse immune reactions [140]. Conclusively, exosomes can be a useful tool for the treatment of oral cancer, but their purification, analysis, and administration are still challenging [145].

## 4. Controlled Drug Delivery Approaches for Oral Cancer

The treatment options for advanced OSCC are limited and suboptimal. Conventional therapeutic approaches (i.e., surgery, chemotherapy, and radiotherapy) significantly impact patient’ wellbeing and quality of life. Thus, there is an imperative requirement for new therapeutic methods with reduced side effects and systemic toxicity. Several controlled drug delivery and release strategies have been developed to overcome the current challenges associated with the parenteral (intravenous, IV) administration of chemotherapeutic agents. These strategies include: the administration of chemotherapeutics via intra-tumoral injection; local delivery; photo-thermal administration using drug-loaded nanoparticles; and ultra-sonoporation using microbubbles (Figure 2). These approaches are reviewed and discussed herein regarding oral cancer.

### 4.1. Intra-Tumoral Drug Delivery in Oral Cancer

One approach is local intra-tumoral administration [146,147]. Li et al. developed a controlled release system that optimized the combined therapeutic benefits of two anticancer drugs while minimizing their side effects, by using suberoylanilide hydroxamic acid (SAHA) and cisplatin (DDP) loaded into PECE hydrogel for the OSCC treatment. Six mice groups were comparatively analyzed (1st group was injected with normal saline (NS); the 2nd was injected with blank hydrogel; the 3rd with SAHA; the 4th with DDP; the 5th with SAHA-DDP; and the 6th with SAHA-DDP/PECE; the mice in the sixth group had the smallest tumor volume with no noticeable systemic cytotoxicity compared to other groups at the end of the study [137]. Intra-tumoral delivery of chemotherapeutic drugs incorporated in a hydrogel is considered as a promising approach for further exploration of OSCC treatment [137].

### 4.2. Local Drug Delivery in Oral Cancer

Local drug delivery is a tumor-targeted approach that delivers the drug to the proximity of the tumor. With this approach, the drugs enter the systemic circulation to a lesser extent compared with other administration routes, thus limiting the adverse side effects of the drugs on healthy cells [148]. For example, locally delivered drugs formulated inside nanoparticles can reach cancer cells passively or through active targeting. In the case of passive targeting, the nanoparticles reach cancer cells by diffusion and enter the cytoplasm by endocytosis, while in the case of active targeting the nanoparticles are functionalized to identify specific receptors on the cancer cell surface resulting in increased drug delivery inside the cancer cell, leaving the majority of the healthy cells unaffected (Figure 3) [149].

Local delivery of anticancer drugs to the oral cavity provides a convenient and safe local administration, with benefit of rapid turnover of the oral mucosa; this allows for a rapid self-repair after given damage and is a significant advantage that helps alleviate the adverse effects caused by long-term local drug delivery [150]. The majority of studies that employed local drug delivery for OSCC treatment used chitosan as a mucoadhesive polymer. To highlight the promise of local delivery in oral cancer treatment, a remarkable study authored by Arulmozhi et al. reported the encapsulation of ellagic acid (EA, an anticancer drug with poor water solubility and oral bioavailability) inside chitosan nanoparticles, which were then evaluated for their therapeutic efficacy in a human oral cancer-derived cell line (i.e., KB cells). The significant cytotoxicity exhibited by the EA nanoparticles suggested that this system has the potential to overcome the limitations of any drug with poor oral bioavailability via targeted local delivery to cancer cells by enhancing its local therapeutic benefits while reducing its systemic side effects [108].

### 4.3. Phototherapy Approaches in Drug Delivery

Phototherapy is a minimally invasive method that is commonly used in the treatment of neoplastic disease. The first phototherapeutic technique is photodynamic therapy (PDT), consisting of administration of a photosensitizing agent followed by irradiation, which is absorbed by the agent at a specific wavelength. The photosensitizer generates reactive oxygen species (ROS) following the utilization of near-infrared (NIR) light, which results in the apoptosis of cancer cells. This process has proven to be efficient in killing the cancer cells, with the limitation that accumulation of the photosensitizer in the tumor is relatively low [151,152]. Photo-thermal therapy (PTT) is another method of phototherapy, which employs light absorbing agents to generate heat, that damages cancer cells and consequently eliminates the tumor [153]. However, PTT is not considered for clinical applications because the laser power density is high and can also damage the surrounding normal tissue [152].

Drug delivery systems (DDS) can improve the phototherapy techniques and address their limitations. Recent studies have focused on incorporating chemotherapeutic agents and photosensitizers or light absorbing agents into nanocarriers. After delivery of these agents at the tumor site, local irradiation has resulted in the killing of the cancerous cells and tumor shrinkage. Current research studies are focused on the use of magnetic nanoparticles for targeting or tracking cancer cells by magnetic resonance imaging (MRI) [154,155,156,157].

He et al. combined photodynamic therapy (PDT) with chemotherapy to simultaneously release anticancer and photosensitizer drugs at the tumor site for the treatment of resistant head and neck cancer. Coordination polymer (NCP)-based core-shell nanoparticles were prepared and loaded with cisplatin and the photosensitizer pyrolipid. They performed in vivo studies, where mice were treated with a combination of nanoparticles loaded with cisplatin and pyrolipid; a remarkable tumor reduction (83%) occurred in cisplatin-resistant SQ20B subcutaneous xenograft murine HNSCC model after the combined treatment of loaded nanoparticles and irradiation. This system of delivery allowed for high loadings of cisplatin and pyrolipid to be locally released after irradiation at the tumor site, with increased anticancer effects as compared to monotherapy [158].

### 4.4. Microbubbles Mediated Ultrasound in Drug Delivery

Microbubbles are micrometer-sized (1–2 µm) gas bubbles that are used as ultrasound contrast agents. The injection of microbubbles into blood circulation improves the contrast of ultrasound images. In addition to their diagnostic usage, the combination of microbubbles and ultrasound can be used in local drug delivery for the treatment of cancer. Microbubbles can be targeted to specific tumor sites by incorporation of ligands or monoclonal antibodies binding to receptors expressed on cancer cell membranes. The combination of chemotherapeutic agents with microbubble-mediated ultrasound therapy increases drug uptake in targeted tissues through so-called ‘sonoporation’, improves the drugs’ biodistribution and decreases their systemic toxicity [159,160,161]. Sonoporation is defined as a drug delivery system that uses ultrasound for intracellular delivery of agents that cannot move into cancerous cells under normal conditions [42].

One crucial strategy for the treatment of HNSCC is the inhibition of EGFR signaling, but current methods cannot suppress this signaling completely. EGFR inhibition can occur through RNA interference by using microbubbles as nucleic acid delivery vectors. Microbubbles delivered to the site get ruptured by ultrasound-targeted microbubble destruction (UTMD) resulting in drug release from the microbubbles’ shell to the insonified area [162].

Recently, Hirabayashi et al. developed anti-EGFR antibody-conjugated microbubbles for colon squamous cell carcinoma treatment. In in vivo studies, anti-EGFR-microbubbles were injected directly into the tumor, while the anticancer drug bleomycin (BLM) was injected via the tail vein. The findings of this study showed that anti-EGFR-microbubbles bound to EGFR on Ca9-22 cells, and the BLM uptake was increased following anti-EGFR-microbubbles binding to cancerous cells. This system is promising to enable effective targeted delivery of anticancer drugs into oral cancer cells [42].

Carson et al. highlighted the potential use of microbubbles as carriers of anti-EGFR siRNA along with ultrasound-targeted microbubble destruction (UTMD) in SCC-VII-induced murine squamous cell carcinoma model. Delivery of microbubbles to the tumor site, where they were ruptured by UTMD and resulted in drug release from the microbubbles to the insonified area, led to tumor growth suppression in mice with OSCC [162]. Recent studies on drug delivery for oral cancer are summarized in Table 2. 

A novel immunotherapy strategy involves using small molecules as monotherapy or combined with other anticancer therapies [163]. The main advantages of these small molecules are good oral bioavailability, ability to penetrate the physiological barriers, precise formulations and dosing options, and lower cost to produce and administer [163,164]. A summary of the small molecules designed for HNSCC and/or OSCC treatment are provided in Table 3.

There is also promising burgeoning research on immunotherapy and gene therapy for oral cancer treatment, and these therapies can also benefit from DDS [162,182,183]. A significant advantage of DDS is their clinical potential for oral cancer diagnostics and treatment simultaneously. Therefore, designing theranostic systems containing both imaging and anticancer agents will significantly improve the diagnosis and treatment of OSCC at early stages.

## 5. Conclusions and Future Perspective

The major challenge in the management of HNSCC patients today is the development of the evasive cancer cell resistance to conventional therapies. Drug delivery systems employed for the administration of chemotherapeutic agents have shown promise in the abilities to overcome the limitations of the conventional anticancer therapeutic approaches. Drug delivery systems for oral cancer consist of three major components: the anticancer agents (single or multiple); carriers to encapsulate the agents; and the methods of delivering the agents to the tumor site. The carriers can be chosen among natural, synthetic, or a combination of materials. They can be prepared in the form of hydrogels or nanocarriers, including nanoparticles and nanolipids. New drug delivery approaches in oral cancer focused on intratumoral or local drug delivery, photothermal therapies combined with DDS, and delivery using ultrasound-mediated microbubbles. Even though controlled drug delivery systems have been around for more than 30 years, improving clinical efficiency and release profiles of anti-cancer drugs as well as lowering their side effects remains a challenge. One of the main hindrances for the commercialization of these systems is the low production reproducibility. Currently, most research investigations are still focused on in vitro or in vivo studies, whereas only a few systems have been implemented into the clinic (Table 2). A nano-formulation of DOX (liposomal-encapsulated formulation of DOX, DOXIL®) was approved by the USA Food and Drug Administration (FDA) in 1995 [184] and is used for breast and ovarian cancer treatment [185]. Similar or novel formulations and delivery methods are required to address unmet needs for the treatment of oral cancer. A personalized, reliable drug delivery system explicitly tailored on the unique genetic, molecular, histological, and circadian profile of a given tumor and a given patient seems the ideal approach in treating patients with oral cancer and beyond.

## Figures and Tables

**Figure 1 pharmaceutics-11-00302-f001:**
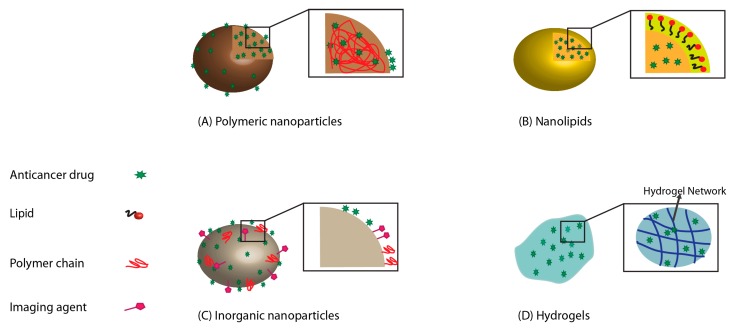
Different carriers used for oral cancer: (**A**) polymeric nanoparticles; (**B**) nanolipids; (**C**) inorganic nanoparticles; (**D**) hydrogels.

**Figure 2 pharmaceutics-11-00302-f002:**
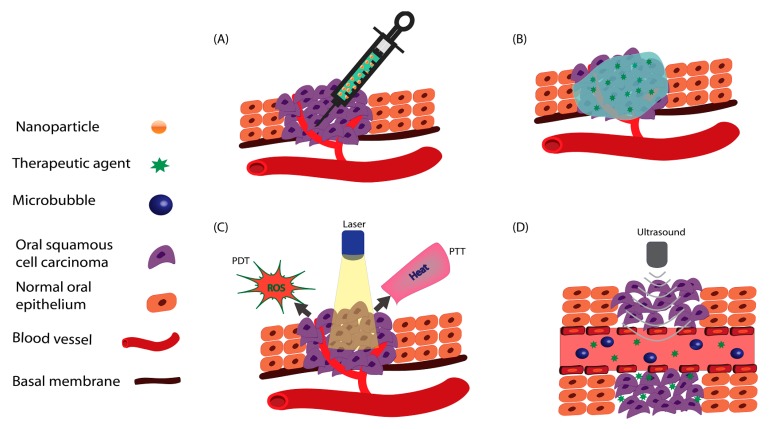
Different controlled drug delivery approaches: (**A**) Intra-tumoral drug delivery; (**B**) local drug delivery; (**C**) photo-thermal therapies combined to drug delivery systems; (**D**) ultrasound-mediated microbubble.

**Figure 3 pharmaceutics-11-00302-f003:**
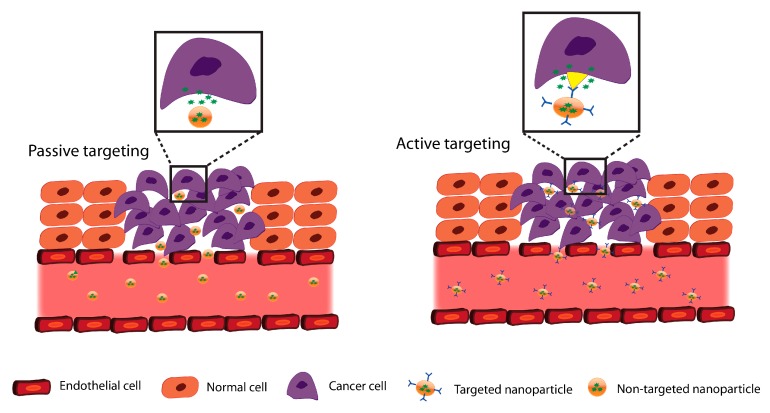
Tumor targeting approaches in oral drug delivery.

**Table 1 pharmaceutics-11-00302-t001:** Carriers for drug delivery in oral cancer treatment.

Carriers for Drug Delivery	Advantages	Disadvantages	References
Polymeric nanoparticles	Biodegradable and biocompatible Suitable for controlled and sustained drugs release with increased therapeutic efficacy and reduced side effects	Difficult to handle due to particle-particle aggregation Cytotoxic after internalization into cells Not suitable for the release of proteins including antibodies Associated with an immune response or local toxicity upon degradation	[82,83,84,85,86]
Inorganic nanoparticles	Target can be site specific by attaching the ligand to the nanoparticle (e.g., magnetic nanoparticles) Higher photostability compared to organic dyes	Toxicity Limited effective delivery due to limited penetration depth for photothermal therapy Cannot deliver biomacromolecules (e.g., proteins)	[87,88,89]
Nanolipids	Highly stable Provide controlled release of drugs to protect them from chemical degradation Encapsulate and deliver drugs with low aqueous solubility Able to penetrate deeply into tumors Suitable for local delivery of anticancer drugs	Crystalline structure provides limited space to accommodate drugs Solid lipid nanoparticles (SLNs) show initial burst drug release Aggregation or gelling of nanostructured lipid carriers (NLCs) during storage Associated with immune response	[83,90,91,92,93]
Hydrogels	Injectable to a specific site Do not dissolve in water at physiological temperature and pH Maintain their structural integrity and elasticity even after retaining large amounts of water High drug loading capacity Ability to deliver hydrophilic and hydrophobic drugs	Poor mechanical properties Difficult to handle Expensive Initial burst	[94,95]

**Table 2 pharmaceutics-11-00302-t002:** Drug delivery studies for the treatment of oral cancer. OSCC: oral squamous cell carcinoma; PLA: poly(lactic acid); SAHA: suberoylanilide hydroxamic acid; DDP: cisplatin; EGFR: epithelial growth factor receptor.

Study	Outcomes	Material	Anticancer Drug/Small Molecules	Target Cells/Target Tumor	Delivery Approach	Type of Study	Sex/Species	Reference
Microbranchytherapy for intratumoral injection of holmium-166 microspheres into 13 cats with inoperable OSCC	Local response rate: 55% Mean survival time: 113 days overall and 296 days for the cases with local response	PLA microspheres loaded with holmium acetylacetonate and then suspended in Pluronic F-68 solution	Holmium-166 microspheres	Tumors located in the: tongue/sublingual (*n* = 10); gingiva of the mandible (*n* = 1); gingiva or the maxilla (*n* = 2)	Intratumoral injection of radioactive agents	In vivo	Eight male and five female cats	[165]
Injection of drug loaded gels into tumors (up to 6 weeks treatments), at dosage: 0.25 mL of active or placebo gel per cm^3^ of the tumor up to 10 mL total	The tumor response noted in 29% of patients, including 19% cases with complete responses in the drug-loaded gel group versus 2% for placebo (*P* < 0.001).	Purified bovine collagen/gel	Cisplatin/Epinephrine	Head and neck tumors	Intratumoral	Clinical study (178 patients pretreated with recurrent or refractory HNSCC); prospective, double-blind placebo-controlled phase III trials	Male and female humans	[147]
SAHA and DDP were loaded into a biodegradable and thermosensitive hydrogel (PECE)	Mice treated with SAHA-DDP/PECE had the smallest tumor volume (62.43 mm^3^) compared to other groups tumor volume.	PECE	Cisplatin (DDP)/SAHA	In vitro: HSC-3 and HOK16-E6E7 cells. In vivo: 2 × 10^6^ HSC-3 cells were injected subcutaneously into the right flank regions	Intratumoral	In vitro and in vivo	Female mice	[137]
Synthesizing DTX encapsulated PLGA nanoparticles for in situ delivery to the tumor site	The slow release profile of the drug (60% of DTX released in 9 days) Higher cytotoxic effect against SCC-9 cells compared to free drug	PLGA	Docetaxel (DTX)	Human tongue squamous carcinoma derived cell line SCC-9	Intratumoral	In vitro	N/A	[166]
Irradiation following intra-tumoral injection of gold nanorods (GNRs) conjugated with rose bengal (RB)	The tumor inhibition rate was significant (95.5%) on the 10^th^ day after treatment for (f).	Gold nanorods (GNRs)/Rose Bengal	-	Tumors induced in hamster cheek pouches	Intratumoral combined with photo-dynamic (PDT) and photothermal (PTT) therapy	In vitro and in vivo	Male hamsters	[167]
Synthesizing and drug encapsulation of EA loaded chitosan nanoparticles	Sustain drug release by 48 h Decreased proliferation of human oral cancer KB cell lines (in vitro)	Chitosan	Ellagic acid (EA)	Human oral cancer KB cell line	local	In vitro	N/A	[108]
Curcumin-loaded in PCL nanoparticles and coated with chitosan as a mucoadhesive polymer	Reduced viability of SCC-9 human oral cancer cell line Decreased toxicity of curcumin incorporated in nanoparticles compared to its free state	Chitosan	Curcumin	SCC-9 human oral squamous carcinoma cell; for permeation studies: esophageal mucosa of at least two different animals	local	In vitro	N/A	[104]
Nano-emulsions loaded with Gen and coated with chitosan in the form of tablets	Controlled release profile Anticancer activity against two oropharyngeal carcinoma-derived cell lines Both formulations showed equivalent cell kill ratio within 48 h	Nanoemulsion, chitosan, cellulose microcrystalline, dextrose	Genistein (Gen)	SCC-4 cells, FaDu cells, and murine connective tissue fibroblasts (L929) (in vitro)/ porcine buccal Mucosa (ex vivo)	local	In vitro and ex vivo	N/A	[168]
Using MTX loaded liposomes to prepare the mucoadhesive film	Increased apoptosis rate in HSC-3 cells by three fold in M-LP-F7 The pro-oxidant effect in HSC-3 cells by M-LP-F7	Liposomes, chitosan (CH), poly(vinyl alcohol) (PVA), hydroxypropyl methylcellulose (HPMC)	Methotrexate (MTX)	HSC-3 cells	local	In vitro	N/A	[169]
Preparation of a targeted nanoparticle platform combing Pc 4 with IO and a cancer targeting ligand, then intravenous injection of non-formulated Pc4 and two nanoparticle formulations: targeted (Fmp-IO-Pc4) and non-targeted (IO-Pc4) were administered to mice	Significant tumor inhibition in both Fmp-IO-Pc4 and IO-Pc4 compared to free Pc4 Significant reduction in tumor volume in targeted nanoparticles (Fmp-IO-Pc 4) compared to IO-Pc4	Iron oxide (IO) nanoparticles	PDT drug (Pc 4)	In vitro: M4E, M4E-15, 686LN, and TU212 cell lines	PDT	In vitro and in vivo	Female mice	[170]
Preparation of gold nanoparticles conjugated with anti-EGFR antibody, then evaluation of the effect of PDT combined with administration of anti-EGFR antibody conjugated Au nanoparticles on two OSCC lines and one epithelial cell line	No photothermal destruction was seen in any of the cell lines in the absence of Au nanoparticles, but one-quarter of this energy was enough to kill the tumor cells in the presence of anti-EGFR/Au nanoparticles	Anti-EGFR antibody conjugated gold nanoparticles	-	Two OSCC cell lines (HSC 313 and HOC 3 Clone 8 ); one benign epithelial cell line (HaCaT)	PDT	In vitro	N/A	[110]
Preparation of self-assembled core-shell nanoparticles loaded with cisplatin and pyrolipid for treatment of resistant head and neck cancers.	Reduced the tumor volume only in NCP@pyrolipid plus irradiation group in cisplatin-resistant SQ20B tumors by 83% No tumor growth inhibition was observed in NCP@pyrolipid without irradiation	1,2-dioleoyl-sn-glycero-3- phosphate sodium salt (DOPA) coated nanoscale coordination polymer (NCP)-based core-shell Nanoparticles with PEG	Cisplatin and pyrolipid (as photosensitizer)	In vitro: cisplatin-sensitive HNSCC135 and SCC61 as well as cisplatin-resistant JSQ3 and SQ20B In vivo: SQ20B subcutaneous xenograft murine models	PDT	In vitro and in vivo	Female Mice	[158]
Injection of anti-EGFR-microbubbles into the tumor site, with intravenous injection of BLM 5 min after microbubble injection	Increased BLM uptake after sonoporation with anti-EGFR-microbubbles The greater anti-tumor effect in anti-EGFR-microbubbles compared to microbubbles alone Improved BLM cytotoxicity in Ca9-22 cells in vitro and in vivo	Liposomes with PEG chains	Bleomycin (BLM)	In vitro: Ca9-22 cells In vivo: Ca9-22 cells injected into the back of mice	Local using microbubbles and ultrasound	In vitro and in vivo	Male Mice	[42]
Sonoporation using microbubbles with anti-EGFR antibody and administration of BLM to assess its effect on Ca9-22 growth	Remarkable inhibition of Ca9-22 cells growth Surface deformation of Ca9-22 after sonoporation in the presence of antibody Increased number of apoptotic cells with using a low dosage of BLM and the Fab fragment of an anti-EGFR antibody	SonoVue as microbubble agent	BLM	Ca9-22 cell line	Local using microbubbles and ultrasound	In vitro	N/A	[43]

**Table 3 pharmaceutics-11-00302-t003:** Monoclonal antibodies-based therapies for the treatment of head and neck cancer.

Drugs	Mechanism of Action	Reference
Cetuximab, panitumumab, zalutumumab and nimotuzumab	EGFR inhibitors	[171]
Gefitinib, erlotinib, lapatinib, afatinib and dacomitinib	EGFR tyrosine kinase inhibitors	[171]
Bevacizumab	VEGF inhibitors	[171]
Sorafenib, sunitinib and vandetanib	VEGFR inhibitors	[171]
Rapamycin, temsirolimus, everolimus, torin1, PP242 and PP30, BYL719	PI3K/AKT/mTOR pathway inhibitors	[171,172]
Pembrolizumab and nivolumab	Anti-PD-1 antibodies	[171]
Motolimond ( VTX-2337)	TLR8 agonist	[173]
AZD1775 (Adavosertib)	Elective small molecule inhibitor of WEE1 G2 checkpoint serin/threoin/protein kinase	[174]
Abemaciclib ( LY2835219)	Cyclin-dependent kinase inhibitor	[175]
TPST-1120	Selective antagonist of PPARα	[176]
Sitravatinib (MGCD516)	RTK inhibitor	[177]
Nintedanib (BIBF1120)	Triple receptor tyrosine kinase inhibitor (PDGFR/FGFR and VEGFR)	[178]
Durvalumab (Imfinzi, MEDI4736)	(IgG1κ) monoclonal antibody	[179,180]
Tremelimumab	Anti-CTLA4 antibody	[170,181]

Abbreviations: EGFR, epidermal growth factor receptor; VEGF, vascular endothelial growth factor; VEGFR, VEGF receptor; PI3K, phosphatidylinositol 3-kinase; AKT, serine/threonine-specific protein kinase; mTOR, mammalian target of rapamycin; PD-1, program death receptor 1; TLR8, a selective toll-like receptor 8; PPARα, peroxisome proliferator-activated receptor alpha; RTK, receptor tyrosine kinase; PDGF-R, Platelet-derived growth factor receptor; CTLA4, cytotoxic T-lymphocyte associated antigen-4; IgG1κ, human immunoglobulin G1 kappa; WEE1, Wee1-like protein kinase.

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
