# Peer review of "Controlled Drug Delivery Systems for Oral Cancer Treatment—Current Status and Future Perspectives"

_pharmaceutics, 2019, doi:10.3390/pharmaceutics11070302_

Round 1

Reviewer 1 Report

The authors present drug delivery systems targeting oral cancer cells in this review article. The contents of the article are well organized and the article would provide fruitful information for broad readers. The reviewer strongly recommend the review article is acceptable in Pharmaceutics. However, the reviewer would like to suggest that the authors should add information on the size, shapes, and other characteristics of (nano)particles used as drug carriers in the literatures cited as references because such information is quite helpful for the development of new drug carriers. 

Author Response

Thank you so much for your insightful comment. The requested information was added in the manuscript which is highlighted in yellow. 1)Lane 329-324
2)Lane 346-354
3)Lane 378-379
4)Lane 394-397
5)Lane 404-414
6)Lane436-438

The revised manuscript has been attached.

Reviewer 2 Report

Ref : Controlled drug delivery systems for oral cancer treatment - current status and future perspectives

This is an interesting paper, well structured, providing and excellent summary of the works done in this area.

I do not have particular remarks and support its publication, except may be to simplify the table 2.. I believe this article will be useful for all readers acting  in this field.

Author Response

Thank you so much for your great comment. Table 2 was simplified.

The manuscript has been attached.

Reviewer 3 Report

Reviewer's report

Title:

Controlled drug delivery systems for oral cancer treatment - current status and future perspectives

Version: 1Date:  2019 June 14th

Reviewer's report:

In this manuscript, the Authors reviewed the current knowledge about the diverse drug delivery methods for oral squamous cell carcinoma treatment. In particular they report recent improvements to render more effective the chemotherapeutic agents administration in a time specific manner, the so called chrono-chemotherapy. For instance, In HNSCC a combination of Docetaxel, Cisplatin, Fluorouracil in the evening had less side effects than morning dosing. 

Overall this review is well written and summarize the most important issues in this field. I have found only some minor point to revise as follows:

Minor Points:

-       Section Introduction, page 2, lane 69: specify OPC meaning (Oropharyngeal cancer), as it appeared for the first time here

-       Section Introduction, page 2, lane 79: the sentence OSCC is the most common second cancer in transplant patients needs a reference.

-       Section Introduction, page 3, lane 130: the acronym DDS has been already addressed in lane 123. The sentence :”In recent years …non-cancerous cells.” Is redundant, please eliminate it.

Author Response

Thank you so much.

OPC was defined (page 2, lane 69)

The reference added (page 2, lane 79,80)

The definition was removed in page 3, lane 130

The sentence:”In recent years …non-cancerous cells. was eliminated (page3, lane 130)

The revised manuscript has been attached.
